# Debugging: Strategies and Considerations for Efficient RNAi-Mediated Control of the Whitefly *Bemisia tabaci*

**DOI:** 10.3390/insects11110723

**Published:** 2020-10-22

**Authors:** Emily A. Shelby, Jeanette B. Moss, Sharon A. Andreason, Alvin M. Simmons, Allen J. Moore, Patricia J. Moore

**Affiliations:** 1Department of Entomology, University of Georgia, Athens, GA 30602, USA; eshelby@uga.edu (E.A.S.); jeanette.moss@uga.edu (J.B.M.); ajmoore@uga.edu (A.J.M.); 2U.S. Department of Agriculture, Agricultural Research Service, U.S. Vegetable laboratory, Charleston, SC 29414, USA; sharon.andreason@usda.gov (S.A.A.); alvin.simmons@usda.gov (A.M.S.)

**Keywords:** whitefly, *Bemisia tabaci*, RNAi, RNA interference, integrated pest management

## Abstract

**Simple Summary:**

The whitefly *Bemisia tabaci* is a crop pest insect that is difficult to control through commercially available methods. Technology that inhibits gene expression is a promising avenue for controlling whiteflies and other pests. While there are resources available to make this method insect specific, and therefore more effective, it is currently being used in a way that targets insects broadly. This broad approach can cause potential harm to the surrounding environment. Here, we discuss considerations for using gene-silencing technology as a pest management strategy for whiteflies in a way that is specific to this pest, which will address short- and long-term issues of sustainability. We also provide a way of selecting target genes based on their roles in the life history of the insect, which will reduce the potential for unintended negative consequences.

**Abstract:**

The whitefly *Bemisia tabaci* is a globally important pest that is difficult to control through insecticides, transgenic crops, and natural enemies. Post-transcriptional gene silencing through RNA interference (RNAi) has shown potential as a pest management strategy against *B. tabaci*. While genomic data and other resources are available to create highly effective customizable pest management strategies with RNAi, current applications do not capitalize on species-specific biology. This lack of specificity has the potential to have substantial ecological impacts. Here, we discuss both short- and long-term considerations for sustainable RNAi pest management strategies for *B. tabaci*, focusing on the need for species specificity incorporating both life history and population genetic considerations. We provide a conceptual framework for selecting sublethal target genes based on their involvement in physiological pathways, which has the greatest potential to ameliorate unintended negative consequences. We suggest that these considerations allow an integrated pest management approach, with fewer negative ecological impacts and reduced likelihood of the evolution of resistant populations.

## 1. Introduction

The whitefly *Bemisia tabaci* (Gennadius) (Hemiptera: Aleyrodidae) is a globally important cryptic species complex that causes billions of U.S. dollars in damages to many crops including vegetable and row crops and therefore presents a major threat to food security [1,2,3]. Over 1000 plant species serve as hosts to *B. tabaci* [4], and crops that are damaged range from common fruits and vegetables such as cassava (*Manihot esculenta*), squash (*Cucurbita* spp.), and tomato (*Solanum lycopersicum*) to row crops such as cotton (*Gassypium* spp.) [3]. Given their mode of feeding, the economic damage caused by *B. tabaci* is extensive because it can be either direct or indirect [5]. Like most phytophagous hemipterans, whiteflies feed from the phloem of plants. This feeding behavior affects crops directly in two ways. First, direct feeding can cause adverse plant response such as chlorotic spots on the leaf surface [6]. Damage may result in stunting and can affect the development of reproductive structures of plants, resulting in reduced crop yield in fruits and vegetables [6]. *Bemisia tabaci* can also cause economic loss without direct impact on the health or vigor of the plant. For example, the diet of sugary sap causes the whiteflies to excrete a sticky substance referred to as honeydew, which coats the plant, results in a buildup of mold, and the economic value of the crops is lowered. This is especially true for cotton and other fibrous plants that are used in textiles [7,8,9]. However, the most significant means by which *B. tabaci* causes economic losses in crops is by transmission of plant viruses, including the devastating begomoviruses [10].

Controlling whiteflies has proven difficult. Not only is it difficult to target whiteflies with contact insecticides as they feed from the bottom of leaves, mounting research has made clear that the practice of using pesticides with broad modes of action is costly both economically and biologically, as such methods lead to indiscriminate losses in non-pest insects including beneficial pollinators or natural enemies that feed on *B. tabaci* [11,12,13]. Moreover, *B. tabaci* has developed resistance to most classes of insecticides [14]. Other pest control strategies, such as the use of *Bacillus thuringiensis* (Bt) transgenic crops, have proven ineffective against whiteflies and other sap-sucking insects due to their lack of sensitivity to Cry toxin proteins [15]. Other methods, such as biological control measures, have exhibited some success, but often are not able to reduce pest densities to levels that avoid economic losses in a short period of time [16] and rely on supplemental insecticide application [17,18]. Thus, new methods of control are needed, especially those that can be specifically targeted to *B. tabaci*.

RNA interference (RNAi) has emerged as a promising method for pest management. This technology offers a mode of action targeted to specific genes, allowing different physiological systems to be modified to control insect growth, development, or feeding behavior. This gene-silencing mechanism takes advantage of an evolutionarily ancient method used by cells to stabilize the genome against attacks from RNA viruses and foreign genetic elements. RNAi uses exogenous double-stranded RNA (dsRNA) to silence gene expression by co-opting cellular defense mechanisms to target preselected genes [19] using dsRNA intermediates, which would not be produced by cells ([20]; Reference [21] presents a useful in-depth description of RNAi mechanisms). Upon entry into a cell, dsRNA is cleaved by the enzyme Dicer into short dsRNAs, called short interfering RNAs (siRNAs). The RNA-induced silencing complex (RISC) then degrades the ‘sense’ strand of the siRNAs, leaving the strand that has a sequence complementary to the target gene, called the ‘antisense’ strand, to be used for silencing. The antisense strand becomes incorporated into the RISC complex and binds to the complementary messenger RNA (mRNA) based on base pair recognition. The mRNA is then targeted for destruction so that no protein is made. The increased availability of insect genomic and transcriptomic data makes RNAi possible for many insects. This makes RNAi especially useful for *B. tabaci* because the genomes for three important cryptic species in the species complex (MEAM1/B, MED/Q, and African Cassava whitefly) are published [22,23,24]. Moreover, a full genome is not required, only information on target gene sequence (for a thorough discussion, see [21]). Because RNAi does not require transformation of the genome, it can be readily applied to non-model species [25].

Researchers have already begun to investigate the viability of deploying RNAi against agricultural pests [26,27,28]. Ongoing discussions on the utility of RNAi as a pest management strategy are centered around perfecting delivery methods such as transgenic plant expression of dsRNA [29,30,31,32], topical applications [33,34], and endosymbiont-mediated delivery [35] and the ability of the dsRNA construct to silence the target gene and its efficacy to be appropriately evaluated through controls [36,37]. Many reviews have identified the most common pitfalls encountered at each stage of application and offer suggestions on how to improve RNAi efficiency, especially in terms of delivery methods [38,39,40,41,42]. This body of literature has allowed the scientific community to anticipate possible problems associated with specific RNAi delivery methods applied to *B. tabaci*. However, conceptual frameworks that explore the scope of RNAi’s functionality in pure biological terms, considering the appropriate target pathways and desired ecological outcomes, are lacking. Advancing these discussions demands closer attention to knowledge of species specificity, off-target effects, and the possibility that the target species will evolve resistance is lacking (discussed in [42]). Moreover, the central issue in the use of RNAi as a control strategy, the ideal target, is rarely discussed in terms of lethality without further evaluation of long-term impact, ecological interactions, or sustainability of this approach. This is especially important for complex recalcitrant pests like *B. tabaci*. Here, using *B. tabaci* as a model, we specifically highlight these considerations for the use of RNAi as a part of an integrated pest management strategy. Our goal is to consider sustainability of long-term use based on the unique features of *B. tabaci* and present a conceptual framework for selecting target genes based on their involvement in specific functional pathways.

## 2. RNAi as a Pest Management Strategy: Need for Greater Specificity

RNAi has been used for over 20 years to evaluate gene function in insects [43,44,45,46], but only recently has its utility in more commercial applications, including pest management, been considered [47,48,49,50,51]. Early studies on coleopteran and lepidopteran pests first demonstrated the potential of RNAi to induce lethality through the knockdown of critical functional genes [36,37]. Since this time, advancements in RNAi technology coupled with the increasing availability of genomic data for non-model insects have expanded the capabilities of researchers to develop more species-specific control mechanisms. Despite this, RNAi for pest management continues to rely on broad stroke approaches and molecular considerations with little attention given to species life history and physiology. Moreover, successful implementation of this new technology remains hindered by familiar pest management issues, including cost efficiency and ecological impacts, that ultimately limit the success of RNAi as a long-term solution. Whiteflies are remarkably well studied among insect pests, and many aspects of successful RNAi implementation in *B. tabaci* have been worked out [29,30,31,52,53,54,55,56,57,58]. According to overarching consensus, the formula for effective RNAi pest management of *B. tabaci* is to (1) select a target gene that is crucial for survival, (2) use a plant-mediated delivery method, and (3) prevent dsRNA degradation. While this workflow is useful for current implementations of RNAi, it only partially addresses the question, “What is successful application of RNAi in pest control?”

We contend that successful RNAi-mediated control strategies should offer an effective and sustainable alternative or supplement to conventional methods. If true, the present considerations must be expanded to include strategies that minimize ecological impacts and maximize long-term efficacy. This sentiment echoes increasing calls for more holistic approaches to RNAi-mediated pest management [58]. For the remainder of this review, we focus on three key considerations that have been underrepresented in studies of RNAi in pest management: ecological impact, evolution of resistance, and choice of functional pathway. Each of these considerations relies fundamentally on an understanding of gene function, which we contend must be prioritized for the development of future, species-specific RNAi control strategies.

## 3. Desired Outcomes: Lethal versus Sublethal Effects

Of primary importance for designing and implementing species-specific RNAi control is the careful consideration of desired effects. When it comes to pest management, the desired effect is most often assumed to be lethality [29,59,60,61]. However, in practice, such blunt-force approaches potentially create more problems than they solve. Below, we expand on these shortcomings and also present the case for substituting *sublethal* forms of RNAi control as practically effective, long-term sustainable alternatives to current approaches. In contrast to conventional approaches that manage lethality through dosage administration, we use the term “sublethal” to refer specifically to the magnitude of outcome predicted when knocking down particular gene pathways. In other words, gene targets are selected based on their additive contributions to total fitness as opposed to their necessity for survival.

### 3.1. Short-Term Considerations: Off-Target Effects and Ecological Impact

The first major issue with RNAi applications that result in lethal outcomes lies in the ecological impact of totally eradicating *B. tabaci* from agroecosystems. One possible outcome of this is that a secondary pest species will increase in population density and be more harmful to the cropping system than *B. tabaci* [62]. A second potentially harmful effect is the disruption of complex community interactions in which whiteflies are involved outside of agricultural ecosystems [63,64]. Finally, managing pests through the use of lethal RNAi strategies means that any off-target effects are likely to be especially detrimental [36,65,66]. One of the ways that RNAi can become a biological hazard is that the dsRNA designed to silence specific target genes aligns, imperfectly or perfectly, to gene regions in another species that share high sequence similarity [67,68]. Such effects become increasingly likely when the targets of RNAi are housekeeping genes with critical functions for survival, as these tend to be highly conserved across arthropods [69,70]. For example, one study observed that RNAi-induced silencing of core metabolic genes (v-ATPases) in a pest beetle species had significant, unintended effects on survival of non-target beetles [36]. Deployed in a mixed agroecosystem, such off-target effects could have devastating consequences.

An alternative to lethal RNAi strategies is to develop a framework that more closely aligns with the goals of integrated pest management. One approach currently being developed for RNAi is based on the use of sublethal methods to control pest populations [71,72,73,74]. Together with integrated approaches that target pests only within controlled, locally affected areas, sublethal RNAi approaches offer a promising solution to the issue of unintended ecological consequences arising from the total eradication of whiteflies. A second major advantage of this approach is that it reduces the severity of unintended effects on non-target species [71,75]. This is true because genes that are less crucial for basic cellular functions are also less likely to be strongly conserved. In addition, in the improbable case of perfect sequence alignment, off target effects are less likely to cause mortality. Further, by using sublethal strategies there is the potential to combine RNAi with more traditional and ecologically-based approaches, including the use of insecticides and natural enemies, which may enhance efficacy at reduced cost [71]. The use of sublethal strategies could make it possible to utilize parental RNAi for long-term control and could reduce the number of treatment applications. However, this strategy may not be feasible for all gene targets, specifically those that affect reproductive fitness.

### 3.2. Long-Term Considerations: Evolution of Resistance

Though the development of resistance is not considered an immediate issue for RNAi-mediated pest control, it remains an important consideration for long-term sustainability [42]. The main way that insects become resistant to RNAi appears to be through the acquisition of more efficient dsRNA degradation capabilities [76]. However, field insect strains that are resistant to RNAi due to an inability to uptake dsRNA in the gut cells have been reported [77]. Such concerns are pronounced when it comes to managing pests with fast generation times.

Whiteflies exhibit very rapid population growth cycles, which are augmented by their ability to flexibly switch between sexual and asexual modes of reproduction [5]. Under optimal conditions, *B. tabaci* females reproduce sexually, resulting in diploid daughters [78] (Figure 1). Because most genetic mutations, including those that confer insecticide resistance, are recessive [79], female offspring must inherit two copies of this new variant to become functionally resistant to RNAi. However, *B. tabaci* is haplodiploid, meaning that females also have the ability to reproduce asexually when suboptimal conditions make it difficult to locate mates [5] (Figure 1). This results in the production of haploid male offspring, which inherit only a single copy of a new variant and therefore always express this new resistance phenotype. Due to their hemizygous condition, males inheriting mutant (resistant) copies are exposed to selection regardless of dominance or recessiveness [79]. The result of this is that can rise to fixation rapidly in populations containing a large proportion of haploid individuals, whereas diploidy delays the evolution of resistance, particularly when beneficial mutations are recessive and their effects are “shielded” in heterozygous states [79,80] (Figure 1B). Hence, systems expressing flexible haplodiploidy are already more likely to evolve resistance to RNAi than systems that rely on strict diploidy for their reproduction [79,80]. Selecting RNAi targets with sublethal effects, as opposed to lethal effects, offers an intuitive path for slowing the evolution of resistance—by relaxing selection against susceptible individuals (Box 1). In the majority of cases where an insect’s pestiferous status can be traced to its high densities, methods of biocontrol that dramatically suppress population growth, rather than inducing mass mortality, are likely to be as effective in mitigating economic damage.

Box 1A population genetics perspective on the evolution of resistance.Imagine a hypothetical scenario in which a random mutation creates a genetic variant for resistance (a “resistant allele”), which initially segregates at low frequencies in the population. Two factors—the magnitude of effect of the allele on an individual’s phenotype and the strength of natural selection on resulting phenotypes—will determine the speed at which a resistance allele rises to fixation in a population. The efficiency of these dynamics is particularly pronounced in instances where control methods cause lethality, as this mimics a phenomenon referred to in nature as ‘hard selection’ [75,81] Hard selection occurs when alleles are inherited as lethal equivalents, causing immediate death of an organism independent of other selection pressures [82]. The result is strong and efficient purging of non-resistant alleles segregating in the population [79]. Consider alternatively a scenario whereby non-resistant alleles do not cause lethality, but instead reduce organismal fitness and gradually suppress population growth. This is referred to as ‘soft selection’ [75,81] and in our example will be applied to any continuous trait imparting sublethal, but severe, effects on fitness. By permitting variation in relative fitness among individuals that carry resistant alleles, soft selection increases the likelihood that some susceptible alleles will be passed on to subsequent generations, thereby relaxing ‘hard selective’ pressures that drive rapid evolution.

## 4. Selecting Suitable Targets Based on Pathway Involvement

To prioritize pathways containing genes that are less highly conserved across arthropods and whose knockdown is less likely to cause lethality, an understanding of species-specific gene function is required. Unfortunately, we do not always have gene function defined across species. Overwhelmingly, given the status of *Drosophila melanogaster* as a long-standing genetic model organism, dipteran model systems serve as the standard from which entomologists have come to understand gene function in insects [83]. Even though genomic and transcriptomic resources are increasingly available for common hemipteran pests, including whiteflies [22,23,24], the function of many insect genes are rarely assigned based on functional genomics or other functional studies. Rather, many of the assigned gene functions have been extrapolated from laboratory experiments on *D. melanogaster* and/or inferred from sequence similarity with *Drosophila* homologues [84,85]. These approaches are imperfect, as hemipteran pests, including whiteflies, exhibit distinct life history strategies and physiology. Thus, current and ongoing functional genomic studies will provide invaluable insight into selecting genes to silence. A further reason that selecting RNAi targets in pathways of known functions is a desirable long-term option over reliance on annotated functions extrapolated from *Drosophila* is that this would allow for more precise scientific monitoring and troubleshooting of RNAi behavior.

Categories of sublethal target pathways include those involved in gametogenesis, immunity, movement, mating, and appetite stimulation. Targeting genes from pathways involved in the detoxification of plant defenses has already shown to be successful at reducing, but not eradicating, *B. tabaci* populations [58]. Table 1 provides examples of potential targets. For *B. tabaci*, genes related to mating behavior may not be effective because of their ability to reproduce parthenogenically. More promising are genes targeting gametogenesis and immunity. Knocking out reproductive capacity is attractive because *B. tabaci* reproduction can be knocked out at several stages. Potential genes include those involved in gamete viability such as *Boule*, *Vasa*, or DNA methyltransferase 1 (*Dnmt1*) [86,87,88,89]. The latter suggestion is a good example of the importance of understanding specific gene function. Conventionally, *Dmnt1* is expected to maintain methylation in the genome. However, methylation in insects does not appear to be universal, and some species such as the genetic model species *D. melanogaster* and *Tribolium confusum* lack *Dmnt1* altogether [90]. Yet in *Oncopeltus fasciatus*, a hemipteran, co-authors have shown through RNAi knockdown that *Dnmt1* has an essential function in gamete production in both males and females and that the silencing of *Dnmt1* results in sterility without causing mortality or disrupting somatic tissue [86,87]. The *Dnmt1* gene sequence has been identified in *B. tabaci* and successfully knocked down using RNAi [91].

Another possible sublethal target is immunity. Targeting immunity genes has the two-fold benefit of reducing fitness and increasing efficiency of the dsRNA construct. The immune system barriers that evolved to protect *B. tabaci* from infection also have the capability to degrade the dsRNA before it can be processed by the endogenous RNAi machinery. In many cases, the dsRNA is ingested and subsequently degraded by nucleases in the saliva or the gut [59]. This phenomenon has been observed in many insects and contributes to variation in RNAi efficiency seen between and among species [100,101,102]. It has been demonstrated in *B. tabaci*, as well as other insect orders, that dsRNA-mediated silencing of digestive nucleases enhances the efficiency of silencing target genes [53,103,104,105].

## 5. Conclusions

The remarkable diversity of pest management strategies is matched only by the ability of insects to overcome them. RNAi strategies that take into consideration life history and population genetics are likely to be valuable control strategies. Success in *B. tabaci* control will depend not only on building on our knowledge of the success and failures of current approaches, but also on how well we address long-i.eterm issues. Using RNAi to resolve these challenges offers a unique opportunity to explore these novel aspects of *B. tabaci*, including evolution of resistance and reproductive biology, while also managing their economic damage. It also makes it possible to address other issues such as application effects and cost issues, which are crucial factors for the development of practical biopesticides. For the best long-term outcomes, future RNAi applications using *B. tabaci* should consider selecting non-lethal target genes that confer continuous traits such as fecundity. Further developing RNAi for use as a sustainable pest management strategy will require extensive collaboration between IPM specialists, laboratory scientists, and public policy makers.

## Figures and Tables

**Figure 1 insects-11-00723-f001:**
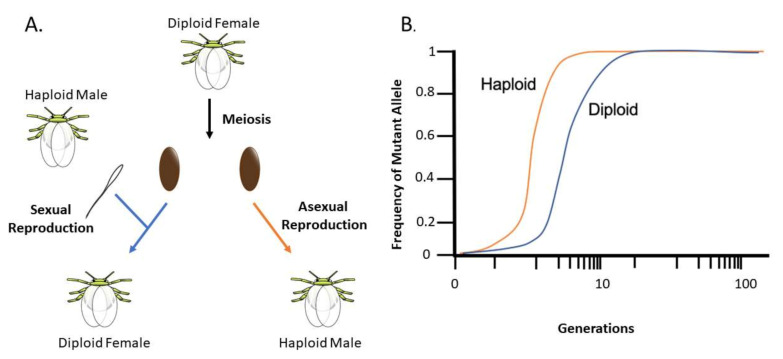
Arrhenotoky in *Bemisia tabaci* and fixation of resistant alleles. (**A**) Sexual and asexual reproduction. Sexual reproduction results in diploid female offspring while asexual reproduction results in haploid male offspring. (**B**) Hypothetical rate of fixation of mutant alleles in haploid vs. diploid organisms. Rate of mutant allele fixation occurs faster in haploid populations (Modeled from [79]).

**Table 1 insects-11-00723-t001:** Examples of potential sublethal genes as targets of RNAi pest control.

Function	Potential Genes	Citations
Appetite stimulation/feeding	*For (foraging)*	[92]
	*Anox (anorexia)*	[93,94]
Immunity/detoxification	*Def (defensin)*	[95]
	*DsRNase*	[53]
	*GST (glutathione S- transferase)*	[58]
Mating/reproduction	*Fru (fruitless)*	[96]
	*Dsf (dissatisfied)*	[97]
	*Croaker*	[98]
	*Per (period)*	[99]
Gametogenesis	*Boule*	[89]
	*Vasa*	[88]
	*Dnmt1 (DNA methyltransferase 1)*	[86,87]

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
