# Peer review of "Debugging: Strategies and Considerations for Efficient RNAi-Mediated Control of the Whitefly Bemisia tabaci"

_insects, 2020, doi:10.3390/insects11110723_

Round 1
Reviewer 1 Report
Shelby and colleagues wrote an excellent review on the use of RNAi to control a pest insect, the whitefly Bemisia tabaci. The manuscript is very useful for thinking about gene silencing strategies that go beyond the target species. The authors were concerned with addressing issues that are often overlooked in the literature, such as ecological impacts and outcomes, evolution of resistance, choice of functional pathway, sustainability and short-/long-term alternative solutions and efficacy.
I present two minor points that I think deserve more attention from the authors. I believe that they can enrich the quality of the manuscript and reach a larger number of readers:
1 - the authors highlighted a lot about the sublethal RNAi strategies but there are no clear examples of what would be considered as lethal versus sublethal. Do they refer to the dose administered, or are there other parameters to differentiate the level of expected and/or observed lethality? In practice, is there a value or percentage or threshold to differentiate the lethal effect from the sublethal effect? Is it possible to predict before the use of RNAi whether the effect will be lethal or sublethal? Please clarify these points in the text.
2 - the literature is rich in the example of parental RNAi knockdown, that is, when dsRNA leads to gene inactivation in offspring embryos. It would be important to discuss this possibility in the manuscript, whether it is useful (explaining utility) or unfeasible (explaining unfeasibility) for the control of Bemisia tabaci.
3 - the manuscript could explore a little more about the usefulness or ineffectiveness of dsRNAs delivery methods, such as topical applications by RNAi spray.
4 - RNAi control experiments, such as the use of GFP and other molecules, are also somewhat controversial in the literature. I believe that mentioning this in the manuscript can alert readers to reflect on everything that needs to be taken into account in the design of gene silencing experiments for the control of insect pests.
Author Response
1 - the authors highlighted a lot about the sublethal RNAi strategies but there are no clear examples of what would be considered as lethal versus sublethal. Do they refer to the dose administered, or are there other parameters to differentiate the level of expected and/or observed lethality? In practice, is there a value or percentage or threshold to differentiate the lethal effect from the sublethal effect? Is it possible to predict before the use of RNAi whether the effect will be lethal or sublethal? Please clarify these points in the text.
We addressed this comment by elaborating on what we mean by “sublethal” (lines 138-142, 144, lines 160-164).
2 - the literature is rich in the example of parental RNAi knockdown, that is, when dsRNA leads to gene inactivation in offspring embryos. It would be important to discuss this possibility in the manuscript, whether it is useful (explaining utility) or unfeasible (explaining unfeasibility) for the control of Bemisia tabaci.
We addressed this comment by suggesting that parental RNAi can be used in sublethal strategies, though it may not be feasible for all genes (lines 169-172).
3 - the manuscript could explore a little more about the usefulness or ineffectiveness of dsRNAs delivery methods, such as topical applications by RNAi spray.
We addressed this comment by rewording statements in lines 91-93 and line 94 to better highlight how the pros and cons of delivery methods have already been discussed in the literature, particularly in the citations provided.
4 - RNAi control experiments, such as the use of GFP and other molecules, are also somewhat controversial in the literature. I believe that mentioning this in the manuscript can alert readers to reflect on everything that needs to be taken into account in the design of gene silencing experiments for the control of insect pests.
We revised lines 90-91 to highlight the ongoing concern of how to evaluate controls.
Reviewer 2 Report
The review entitled “Debugging: Strategies and Considerations for Efficient RNAi-mediated Control of the Whitefly Bemisia tabaci” is well written and brings up several important points, including the fact that insects will eventually overcome whatever control strategy we throw at them, and that RNAi targets that generate sublethal effects could be very valuable. There are however a number of issues I feel need to be addressed before publication of this review.
The largest issue is in section 3.2 where the authors address the evolution of resistance. Here they take an in-depth look at the scenario where resistance is recessive (R= Resistance allele, S= susceptible allele). Since resistance is recessive, RS females are susceptible. These females are shown in Fig 1A producing eggs that are fertilized by R males, BUT not by S males, despite the fact that the RS female shown in Fig. 1B can produce both R- and S-bearing males. Not only is the fact not shown in Fig 1, but lines 178 and 179 state that RS females ONLY produce R males. If the authors are trying to say that all S-bearing males die due to the control measure, then the heterozygous RS female would also be dead since RS females are susceptible (i.e. resistance is recessive). In addition to this oversight, there seems to be a number of errors in the figure. For example, at the bottom of the figure it says RR (Resistant) --- Diploid Females ---- RS (Resistant). Again, if resistance is recessive only RR females, and R males will be resistant to RNAi, and RS will be susceptible. I’m also having a problem with the logic that a haploid/diploid life history will increase the odds of resistance. Will the frequency of the R and S alleles in the population really be significantly altered by this life history trait given the fact that the S allele is “protected” in RS females?
Another issue in section 3.2 is the author’s statement about it being unclear whether field-evolved resistance in western corn rootworm was due to the development of resistance to the target sequence, or dsRNA uptake. Khajuria et al., (ref 77) clearly show AND state that their “observations indicate that the basic siRNA biogenesis machinery is not affected in dsRNA-resistant insects, but rather that dsRNA uptake is impaired in insect gut cells and is the primary cause of resistance to ingested dsRNA.” Given this discrepancy, I feel the authors must revise their wording to reflect what Khajuria et al published, specifically, that resistance is due to loss of dsRNA uptake.
My next points are both a little picky: First the authors imply the need for genomic data - lines 79 to 83 - or at least genomic data for the target gene. However, RNA-Seq data actually fuels much of the RNAi research performed in non-model insects. Therefore I would be much happier if RNA-Seq was included in the list since it is as a rich resource for identification of target genes. Second, lines 197 – 198 state that published insect genomes are rarely fully annotated. My problem isn’t with what the authors are trying to say, but with the terminology. We do indeed “annotate” genomes based on sequence homology with Drosophila and others rather than through “functional genomics” where we are testing each sequence for function. And yes, conserved sequence does not mean conserved function. However, we can “fully annotate a genome” (i.e. this is all sequence based), and still know little about gene function since this requires functional studies (RNAi, CRISPR etc). Therefore my suggestion is to change the wording to reflect that what we lack is functional studies, which are needed if we are to know if conserved sequence s have conserved functions.
Author Response
1 - The largest issue is in section 3.2 where the authors address the evolution of resistance. Here they take an in-depth look at the scenario where resistance is recessive (R= Resistance allele, S= susceptible allele). Since resistance is recessive, RS females are susceptible. These females are shown in Fig 1A producing eggs that are fertilized by R males, BUT not by S males, despite the fact that the RS female shown in Fig. 1B can produce both R- and S-bearing males. Not only is the fact not shown in Fig 1, but lines 178 and 179 state that RS females ONLY produce R males. If the authors are trying to say that all S-bearing males die due to the control measure, then the heterozygous RS female would also be dead since RS females are susceptible (i.e. resistance is recessive). In addition to this oversight, there seems to be a number of errors in the figure. For example, at the bottom of the figure it says RR (Resistant) --- Diploid Females ---- RS (Resistant). Again, if resistance is recessive only RR females, and R males will be resistant to RNAi, and RS will be susceptible. I’m also having a problem with the logic that a haploid/diploid life history will increase the odds of resistance. Will the frequency of the R and S alleles in the population really be significantly altered by this life history trait given the fact that the S allele is “protected” in RS females?
We apologize for the confusion – it was not our intent to develop a complete single gene resistance/susceptibility model. We simply used R and S as labels, and this caused confusion. Therefore, we removed R & S from the figure (Figure 1). The figure was changed to better illustrate arrhenotoky and how these modes of reproduction relate to development of resistance in populations. The figure legend was updated to reflect the changes made to Figure 1 (lines 204-208). Information was also added to explain how the haplodiploid state increases the chances of resistance developing (lines 186-192).
2 - Another issue in section 3.2 is the author’s statement about it being unclear whether field-evolved resistance in western corn rootworm was due to the development of resistance to the target sequence, or dsRNA uptake. Khajuria et al., (ref 77) clearly show AND state that their “observations indicate that the basic siRNA biogenesis machinery is not affected in dsRNA-resistant insects, but rather that dsRNA uptake is impaired in insect gut cells and is the primary cause of resistance to ingested dsRNA.” Given this discrepancy, I feel the authors must revise their wording to reflect what Khajuria et al published, specifically, that resistance is due to loss of dsRNA uptake.
Thank you for the correction. We revised the sentence (line 177-178) per the reviewer’s recommendations.
3 -My next points are both a little picky: First the authors imply the need for genomic data - lines 79 to 83 - or at least genomic data for the target gene. However, RNA-Seq data actually fuels much of the RNAi research performed in non-model insects. Therefore I would be much happier if RNA-Seq was included in the list since it is as a rich resource for identification of target genes.
This is a good point. We now include the mention of transcriptomes (line 79-80).
4 - Second, lines 197 – 198 state that published insect genomes are rarely fully annotated. My problem isn’t with what the authors are trying to say, but with the terminology. We do indeed “annotate” genomes based on sequence homology with Drosophila and others rather than through “functional genomics” where we are testing each sequence for function. And yes, conserved sequence does not mean conserved function. However, we can “fully annotate a genome” (i.e. this is all sequence based), and still know little about gene function since this requires functional studies (RNAi, CRISPR etc). Therefore my suggestion is to change the wording to reflect that what we lack is functional studies, which are needed if we are to know if conserved sequence s have conserved functions.
This is another good suggestion. We reworded the statements in lines 216-218 to clarify as suggested.